# Progress on Genetic Basis of Primary Aldosteronism

**DOI:** 10.3390/biomedicines9111708

**Published:** 2021-11-17

**Authors:** Izabela Karwacka, Łukasz Obołończyk, Sonia Kaniuka-Jakubowska, Michał Bohdan, Krzysztof Sworczak

**Affiliations:** 1Department of Endocrinology and Internal Medicine, Medical University of Gdańsk, 80-952 Gdańsk, Poland; przepona@wp.pl (Ł.O.); sonia.kaniuka@gumed.edu.pl (S.K.-J.); ksworczak@gumed.edu.pl (K.S.); 2First Department of Cardiology, Medical University of Gdansk, 80-952 Gdańsk, Poland; michal.bohdan@gmail.com

**Keywords:** primary aldosteronism, familial hyperaldosteronism, ion channels, aldosterone-producing adenoma, *KCNJ5*, *ATP1A1*, *CACNA*

## Abstract

Primary aldosteronism (PA) is a heterogeneous group of disorders caused by the autonomous overproduction of aldosterone with simultaneous suppression of plasma renin activity (PRA). It is considered to be the most common endocrine cause of secondary arterial hypertension (HT) and is associated with a high rate of cardiovascular complications. PA is most often caused by a bilateral adrenal hyperplasia (BAH) or aldosterone-producing adenoma (APA); rarer causes of PA include genetic disorders of steroidogenesis (familial hyperaldosteronism (FA) type I, II, III and IV), aldosterone-producing adrenocortical carcinoma, and ectopic aldosterone-producing tumors. Over the last few years, significant progress has been made towards understanding the genetic basis of PA, classifying it as a channelopathy. Recently, a growing body of clinical evidence suggests that mutations in ion channels appear to be the major cause of aldosterone-producing adenomas, and several mutations within the ion channel encoding genes have been identified. Somatic mutations in four genes (*KCNJ5*, *ATP1A1*, *ATP2B3* and *CACNA1D*) have been identified in nearly 60% of the sporadic APAs, while germline mutations in *KCNJ5* and *CACNA1H* have been reported in different subtypes of familial hyperaldosteronism. These new insights into the molecular mechanisms underlying PA may be associated with potential implications for diagnosis and therapy.

## 1. Introduction

Primary aldosteronism (PA) is a heterogeneous group of disorders caused by autonomous overproduction of aldosterone with simultaneous suppression of plasma renin activity (PRA). It is considered to be the most common endocrine cause of secondary arterial hypertension (HT) and is associated with a high rate of cardiovascular complications [1]. Most authors believe that PA occurs in about 2–13% of patients with HT, and in the group with HT resistant to pharmacotherapy the percentage of patients with PA may be as high as 20%. Recently, two large studies have been published assessing the incidence of PA in patients with HT. Piaditis et al. estimated that PA occurs in 4.6–16.6% of patients with prehypertension or HT [2]. In another large analysis, Kayser et al. estimated the prevalence of PA at 3.2–12.7% in primary care and 1–29.8% in specialist care [3]. Despite the large discrepancies between individual studies, it should be emphasized that PA is relatively common, especially in patients with more severe forms of HT including resistant HT. It is worth noting that the higher the blood arterial pressure, the higher the prevalence of PA. PA is most often caused by bilateral adrenal hyperplasia (BAH) or aldosterone-producing adenoma (APA). Rarer causes of PA include genetic disorders of steroidogenesis (familial hyperaldosteronism (FA) type I, II, III and IV), aldosterone-producing adrenocortical carcinoma, and ectopic aldosterone-producing tumors [4].

Aldosterone is a steroid hormone produced by the glomerulosa layer of the adrenal cortex. It is the most potent mineralocorticoid hormone regulating the water-mineral balance of the body, which it does mainly through the renin–angiotensin II–aldosterone (RAA) system. The main physiological activators of aldosterone synthesis are angiotensin II, hyperkalemia, hyponatremia, and adrenocorticotropic hormone (ACTH) as well as reduction of circulating blood volume, decrease in blood pressure and renal blood supply [5]. The most important site of action of aldosterone is the distal convoluted tubule, where it binds to the cytosolic mineralocorticoid receptor, modulating the activity of the epithelial sodium and potassium channels. The effect of this modulation is an increase in sodium reabsorption through the renal tubules and an increase in urinary potassium excretion, which in turn leads to changes in osmolality that drive water reabsorption resulting in volume expansion, glomerular hyperfiltration, and suppression of renin and angiotensin II [6]. Suppression of angiotensin II results in greater sodium delivery to the distal nephron, thereby amplifying the aldosterone-driven sodium reabsorption and volume expansion, as well as potassium and acid excretion [5]. The above-described mechanisms explain why patients with primary aldosteronism classically present with hypertension, hypokalemia, and metabolic alkalosis.

Clinically, PA manifests as HT resistant to antihypertensive therapy, often severe, accompanied by varied symptoms including muscle weakness, polyuria, increased thirst, paresthesia and muscle spasms, tetany (symptoms of severe potassium deficiency and alkalosis), normovolemia in the initial period and hypervolemia due to sodium and water retention followed by spontaneous diuresis and normalization of extracellular fluid volume (the “escape” phenomenon”), which is probably associated with increased secretion of atrial natriuretic peptide (ANP). Excessive aldosterone causes necrosis, fibrosis and proliferation of myocytes, myocardial hypertrophy, remodeling and fibrosis of blood vessels, and vascular endothelial dysfunction. In addition, excessive aldosterone adversely affects the kidneys, leading to damage to medium and small arterioles and the development of nephropathy, especially with increased dietary sodium intake. [1,7]. It has been reported that autonomous aldosterone secretion as a subclinical form of PA accompanied by normotension increases the risk of cardiovascular disease, kidney disease, and osteoporosis [8].

Screening for PA is especially recommended in the group of patients with moderate (>160–179/100–109 mmHg), severe (>180/110 mmHg), or resistant HT (>140/90 mmHg despite the use of three antihypertensive drugs) as well as those with idiopathic or diuretic-induced hypokalemia, with concomitant obstructive sleep apnea, or with a randomly detected adrenal tumor (Table 1) [5]. Family history should also be considered, particularly if first-degree relatives have been diagnosed with PA or if there is a family history of HT or cerebrovascular accidents at an early age (<40 years) [9].

It is essential that, before assessing the levels of RAA system hormones, any potassium deficiency is corrected with potassium supplementation in order to ensure its serum level at >4 mmol/L [11]. In addition, drugs that affect the sodium–potassium balance and the RAA system should be discontinued: spironolactone, eplerenone, amiloride and other diuretics (false negative results) four weeks before the examination; beta-blockers, clonidine, methyldopa (false positive results) and dihydropyridine calcium channel blockers, angiotensin-converting enzyme inhibitors (ACEI), angiotensin receptor blockers (ARB) and renin inhibitors (false negative results) two weeks before the examination. The only drugs that, according to many authors, can be used during hormonal testing for PA are verapamil (sustained-release) and alpha-blockers. If high blood pressure requires the use of other antihypertensive drugs, this should be considered when interpreting the test results [9].

The most important findings of laboratory tests that confirm the diagnosis of PA are elevated or non-suppressed blood levels and urinary excretion of aldosterone, low non-stimulated PRA index, and high aldosterone–PRA ratio (ARR), the ratio of blood aldosterone level to PRA [12]. ARR is a good screening test for PA, and additionally, most antihypertensive drugs (apart from aldosterone antagonists and beta-blockers) have little effect on its value. High ARR values (>20–30 ng/dL) are suggestive of PA. In patients with laboratory tests showing an elevated aldosterone level >416 pmol/L (15 ng/dL), decreased PRA < 0.77 nmol/L/h (1 ng/mL/h) and increased ARR, functional tests should be performed (beginning with the aldosterone suppression test); this does not apply to patients with hypokalemia for no other tangible reason or with undetectable direct renin concentration (DRC) and aldosterone concentration >20 ng/dL, who can undergo imaging tests and then be referred for surgery without the need for functional tests to confirm this diagnosis [9].

The dynamic confirmatory tests allow the final diagnosis to be made when there is no suppression or insufficient suppression of aldosterone secretion by physiological stimuli or drugs (fludrocortisone, captopril) and also no increase in PRA after upright test [5]. The physiological stimuli that are used in diagnosis of PA are high-sodium diet and infusion of 0.9% NaCl (saline infusion test). From a practical point of view, the most useful test to confirm PA is the saline infusion test performed in a sitting position. However, a threshold plasma aldosterone level indicative of PA has not been established. So far, values >10 ng/dL have been most often regarded as confirming PA, and <5 ng/dL—as excluding PA.

After the diagnosis of PA is made, diagnostic imaging should be performed in order to determine the cause of the autonomic secretion of aldosterone. In patients with PA, a radiographic evaluation of the adrenal glands by computed tomography (CT) or magnetic resonance (MR) of the abdominal cavity is necessary [13]. In doubtful cases, adrenal scintigraphy is performed in specialized centers (the gold standard in the differentiation of PA forms). Adrenal vein sampling can also be performed to measure aldosterone and PRA levels in the venous blood drained from the adrenal glands [4].

Genetic testing is not a routine part of the diagnosis of PA. PA caused by identifiable germline mutations is rare; familial forms of PA are also extremely rare. The guidelines recommend diagnostic testing for FA type I (so-called glucocorticoid remediable aldosteronism, or GRA) in patients diagnosed with PA at an early age (<20 years) and in those with a family history of PA or cerebrovascular events before the age of 40 [10]. Currently, genetic tests are ordered to detect underlying mutations in the *CYP11B2* and *CYP11B1* genes, replacing the previous method of measuring 18-oxocortisol and 18-hydroxycortisol in urine. In very young patients with PA, it is recommended to test for germline mutation in the *KCNJ5* gene causing FA type III [14]; a detailed discussion of the genetic aspects of PA is provided below.

The diagnosis of PA is important for patients because of the increased risk of cardiovascular disease. The unfavorable effects of aldosterone, in addition to increasing blood pressure, include the stimulation of collagen production, as well as pro-inflammatory and pro-thrombotic effects which increase the remodeling and fibrosis of the myocardium and blood vessels [15]. Patients with PA have a 30% higher risk of stroke, myocardial infarction, heart failure, atrial fibrillation and other arrhythmias, and of metabolic syndrome, than patients with essential HT [1]. These observations, as well as the beneficial effects of causal treatment, confirm the importance and necessity of the diagnosis and treatment of PA.

The guidelines recommend unilateral laparoscopic adrenalectomy in patients with documented unilateral form of the disease (i.e., with APA or unilateral adrenal hyperplasia), unless the patient has contraindications to such treatment [5]. Otherwise, chronic mineralocorticoid receptor antagonists (MRAs) should be used. Pharmacological treatment is also recommended in patients with a positive screening test result, in those who do not consent to further diagnostic tests, or when these are contraindicated. Due to the shorter hospitalization time and the lower incidence of complications, laparoscopic adrenalectomy by an experienced surgeon is recommended. Usually, potassium supplementation needs to be discontinued and antihypertensive treatment reduced in the first days after surgery. Fewer than 5% of patients with PA after adrenalectomy require fludrocortisone treatment due to persistent hypoaldosteronism. Surgical treatment of APA leads to complete remission of symptoms in 35–70% of cases. If the disease is not diagnosed or if it is not treated properly, excess aldosterone, especially with concomitant high salt intake, can not only cause hypokalemia and HT but also has direct adverse effects on the heart and vessels, and may lead to nephropathy [16].

In patients with PA, the risk of cardiovascular complications is higher than in patients with essential HT, a consequence of long-term exposure of the arterial walls to increased blood pressure as well as of the harmful effects of excess aldosterone itself. Both adrenalectomy and pharmacological treatment with MRAs reduce this risk and to some extent reverse the existing complications, although the advantage of surgery over long-term pharmacotherapy has not been clearly proven.

## 2. Genetic Basis of PA

PA caused by identifiable germline mutations is rare. Over the last few years, significant progress has been made towards understanding the genetic basis of PA, classifying it as a channelopathy. Recently, a growing body of clinical evidence suggests that mutations in ion channels appear to be the major cause of APAs, and several mutations within the ion channel encoding genes have been identified. Somatic mutations in four genes (*KCNJ5*, *ATP1A1*, *ATP2B3* and *CACNA1D*) have been identified in nearly 60% of sporadic APAs, while germline mutations in *KCNJ5* and *CACNA1H* have been reported in different subtypes of FA (Table 2).

### 2.1. Familial Hyperaldosteronism

Familial hyperaldosteronism (FA) has a distinct clinical course and treatment. In recent years a significant breakthrough in the understanding of FA has taken place thanks to dynamic technological development and genetic research. The currently-available genetic engineering techniques have allowed the identification of further types of FA, as well as a better understanding of the pathomechanisms underlying these diseases and their implications for therapy.

#### 2.1.1. FA Type I

FA type I, also known as glucocorticoid remedial aldosteronism (GRA), is a rare form of hyperaldosteronism. The prevalence of this syndrome is estimated at less than 1% of PA patients [17]. It is a genetically determined syndrome inherited in an autosomal dominant manner, and is caused by a mutation of two genes: *CYP11B2* (the gene encoding aldosterone synthase) and *CYP11B1* (the gene responsible for 11-hydroxylase that converts 11-deoxycortisol into cortisol) [18]. Aldosterone synthase is the rate-limiting enzyme for aldosterone biosynthesis. There is a crossing-over between the *CYP11B1* promoter and the *CYP11B2* coding region [19]. This produces a hybrid, 11-hydroxylase-synthase, that is capable of synthesizing aldosterone but is influenced by ACTH. It also has the ability to form 18-oxy- and 18-hydroxycortisol (so-called hybrid steroids) from cortisol and 11-deoxycortisol. Since the activity of this hybrid enzyme depends on ACTH, administration of glucocorticoids suppresses the production of aldosterone [18]. All these processes lead to the development of BAH.

FA type I is characterized by the development of HT in childhood or adolescence (<20 years), which is often severe, with frequent organ complications, especially hemorrhagic stroke and ruptured intracranial aneurysms, as well as a higher risk of pre-eclampsia and pregnancy-aggravated HT [18]. Typically, GRA patients do not present hypokalemia; it is most likely related to disturbed circadian rhythm of ACTH secretion [17].

It should be noted, however, that the clinical picture may differ between carriers of the mutated gene within one family. Normal blood pressure values were found in some mutation carriers despite the observed excess aldosterone secretion [20]. Some authors have found a milder course of the disease in women and in patients with lower levels of aldosterone or 18-hydroxycortisol (as its concentration is considered to be the best indicator of chimeric enzyme activity) [21]. The level of blood pressure and the course of HT may also depend on the crossing-over site of the genes encoding the hybrid enzyme. It is presumed that mutations and polymorphisms of other genes responsible for blood pressure regulation may also play a role in the clinical manifestation of this syndrome [22].

FA type I should be suspected in patients with:-PA diagnosed at a young age (before the age of 30);-A positive family history of PA;-Family members with a history of cerebrovascular accidents at an early age [13].

In addition to the tests typically used to detect PA, laboratory diagnostics include the dexamethasone suppression test [13]. With dexamethasone suppression testing, dexamethasone can be prescribed at a dose of 1 mg twice daily for three days. Any serum aldosterone on day three of <4 ng/dL is considered a positive test for GRA. It should be added that the dexamethasone suppression test is not sensitive enough, with a high percentage of false-positive diagnoses, and the final diagnosis of this syndrome requires genetic testing. Genetic testing overcomes these difficulties, requiring a single blood collection. The most widely used test is a long polymerase chain reaction (PCR)-based method. A positive long PCR should usually be confirmed by a Southern blot test.

Once FA is confirmed, the treatment of choice consists of low-dose glucocorticoids to suppress ACTH. Dexamethasone 0.125–0.25 mg/day or prednisone 2.5–5 mg/day has been shown to be sufficient to normalize plasma potassium levels and achieve long-term control of HT without cardiovascular complications [14]. In cases of insufficient blood pressure control, MRAs (spironolactone or eplerenone) or amiloride treatment should be included. In children, eplerenone is preferable in order to avoid the side effects of glucocorticoids or spironolactone.

#### 2.1.2. FA Type II

Another autosomal dominant form of PA is FA type II. According to some authors, this form occurs more often than FA type I due to the fact that a group of patients with uncertain status may include patients with low-renin HT, which in some cases can evolve into fully-symptomatic PA [17]. The main symptoms relate to HT and hypokalemia [23]. Both APA and BAH may develop in the course of this syndrome [18]. Excessive aldosterone production is not suppressed after administration of glucocorticoids, and the *CYP11B1*/*CYP11B2* hybrid mutation has not been found. Some authors express the view that type II FA may be associated with mutations in many different genes, which may be reflected in a very diverse clinical picture [17]. In FA type II, the adrenal glands exhibit similar clinical, biochemical, hormonal and morphological changes (APA or BAH) as in non-genetically determined PA [22]. It is currently believed that the diagnosis of FA type II can be confirmed on the basis of:-Finding FA in at least two relatives of a PA patient;-Exclusion of type I FA.

Confirmation of FA type II is based on typical PA laboratory tests and positive dynamic confirmatory tests as well as the absence of the chimeric gene responsible for FA type I.

The results of the research carried out so far show that the development of this syndrome is caused by a mutation of an as yet unidentified gene with a locus at chromosome 7p22 [24]. No evidence of single-nucleotide polymorphism copy number variation between pairs was found. In addition, large genomic deletions or insertions at 7p22 were excluded and the candidate gene list for this locus was refined; however, the mutations causing FA type II have remained elusive. The Brisbane group closely analyzed and hunted for the gene(s) responsible for FA type II [25]. The genotypes of five pedigrees were analyzed using seven closely spaced microsatellite markers at 7p22, and two-point and multipoint logarithm of odds (LOD) scores were calculated in order to assess linkage with FA type II. Linkage studies have excluded linkage with:-*AT1* encoding type 1 angiotensin II receptor isoform, which has a vasopressor effect and regulates aldosterone secretion;-*CYP11B2* encoding aldosterone synthase;-*MEN1* encoding menin 1, which functions as a transcriptional regulator;-*RBaK* encoding RB-associated KRAB zinc finger protein; this gene encodes a nuclear protein which interacts with the tumor suppressor retinoblastoma 1 (this protein contains a Kruppel-associated box (KRAB), which is a transcriptional repressor motif);-*PMS2* encoding mismatch repair endonuclease PMS2 (Postmeiotic Segregation Increased 2);-*GNA12* encoding guanine nucleotide-binding protein (G protein) subunit alpha 12, which is involved as a modulator or transducer in various transmembrane signaling systems.

The combined multipoint LOD score for three analyzed families showed linkage at 7p22 which was highly significant for markers D7S462 and D7S517. The results of the study support the notion that FA type II may be genetically heterogeneous.

The results of recent research indicate that chloride channel (ClC) mutations may play an important role in the genetic basis of PA. ClC-2, the voltage-gated chloride channel encoded by *CLCN2*, is expressed in almost all tissues and may have roles in ion homeostasis and transepithelial transport. ClC-2 is found in brain, kidney, muscle, lung and intestine tissues [26]. Despite widespread *CLCN2* expression, subjects with gain-of-function *CLCN2* variants shared no apparent pathology other than PA, whereas loss-of-function *CLCN2* variants caused slowly developing leukoencephalopathy with ataxia, blindness and male infertility [26,27].

The *CLCN2* RNA is also found in the adrenal gland. In zona glomerulosa cells, ACTH-activated Cl-currents have been described; however their outward rectification sets them apart from hyperpolarization-activated ClC-2 currents. The ClC-2 channels expressed in adrenal glomerulosa open at hyperpolarized membrane potentials. Channel opening depolarizes glomerulosa cells and induces expression of aldosterone synthase. Mutant channels show gain of function, with higher open probabilities at the glomerulosa resting potential [28]. The data indicate that *CLCN2* mutations cause PA.

Two recent studies demonstrated the discovery of mutations in the *CLCN2* chloride channel causing inheritable PA, which may account for at least some cases of FA type II. Fernandes-Rosa et al. performed whole-exome sequencing in patients with early-onset PA and identified de novo heterozygous glycine (Gly) and asparagine (Asp) mutation in the *CLCN2* gene (c.71G>A/p.Gly24Asp) localized in its N-terminal cytoplasmic domain [28]. The *CLCN2* p.Gly24Asp mutation is located in a highly conserved ‘inactivation domain’ 2,3 of the channel. Deletions and point mutations in this region and an intracellular loop 2 lead to ‘open’ ClC-2 channels that have lost their sensitivity to voltage, cell swelling, and external pH. Likewise, insertion of the p.Gly24Asp mutation drastically changed voltage-dependent gating of ClC-2 and dramatically increased Cl- conductance at resting potentials, leading to depolarization of the zona glomerulosa cell membrane and thereby opening the voltage-gated calcium2+ (Ca^2+^) channels which trigger autonomous aldosterone production by increasing intracellular Ca^2+^ concentrations. Moreover, the increased Cl- currents may overcome the hyperpolarizing currents of potassium (K^+^) channels that normally determine the glomerulosa cell resting potential. The inhibition of these K^+^ channels upon, e.g., angiotensin II stimulation, or the depolarizing currents mediated by these channels upon increases in extracellular K^+^, are the main mechanisms triggering aldosterone production under physiological conditions [26].

Sholl et al. analyzed a multiplex family with FA type II and 80 additional probands with unsolved early-onset PA and performed exome sequencing on three affected subjects, identifying two shared novel protein-changing heterozygous variants in *CLCN2* [27]. Novel heterozygous variants in *CLCN2* were found among eight probands, including two de novo mutations and four independent occurrences of a mutation encoding an identical p.Arg172Gly substitution. Moreover, all relatives with early-onset primary aldosteronism carried the *CLCN2* variant found in the proband.

#### 2.1.3. FA Type III

Recently, a new form of FA was described (FA type III) characterized by severe HT in early childhood and associated with marked aldosteronism, hypokalemia, and significant target organ damage [29]. FA type III is exceedingly rare and presents with enormous production of 18-oxy- and 18-hydroxycortisol resulting from the hybrid *CYP11B2*/*CYP11B1* enzyme (similarly to FA type I). It is currently known that the main genetic background includes mutations that implicate gain-of-function mutations of the *KCNJ5* (potassium voltage-gated channel subfamily J member 5) gene coding for GIRK4 (G-protein-activated inward rectifier potassium channel 4) [30]. To date, six *KCNJ5* germline mutations associated with FA type III have been reported for a total of 12 families and 22 affected family members [31].

In FH type III, the responses of aldosterone and cortisol to the dexamethasone suppression test are markedly altered. In fact, when the test was performed, FA type III patients displayed a paradoxical increase in aldosterone to twice the baseline level and a lack of suppression of cortisol levels, which despite being within the normal range, indicated defective regulation and inappropriate production [32]. These clinical and biochemical features are more severe than those displayed by families with GRA and distinctly different from those of FA type II, where the onset of the disease is usually in adulthood. Affected FA type III patients not only displayed particularly high aldosterone production but also a poor response to full doses of several classes of antihypertensive drugs, including spironolactone and amiloride; this distinguishes FA type III from the other familial forms and sporadic PA, because spironolactone is usually successful in controlling blood pressure [23]. The hyperaldosteronism is caused by massive diffuse BAH. Due to ineffective aggressive antihypertensive therapy and BAH, the treatment of choice is bilateral adrenalectomy to control blood pressure [14].

FA type III was described in a study by Geller et al.; however, the authors could not identify the genetic alteration responsible for the disease [29].,They were able to exclude the involvement of potential candidate genes such as *CYP11B1* and *CYP11B2*, the angiotensin II and ACTH receptors, DAX-1 (dosage-sensitive sex reversal, encoded by the *NR0B1* gene), which has been implicated in congenital adrenal hypoplasia, Ad4BP, a zinc finger DNA-binding protein and transcription factor essential for the transcription of steroidogenic p450 genes, and the nerve growth factor inducible factor B family members of nuclear receptor Nur77 and Nur1, which have been implicated in adrenal zonation and potentially involved in the regulation of aldosterone synthase.

The recent discovery of the KCNJ5 mutation as the etiology of FA type III has allowed understanding of the pathomechanism of hyperaldosteronism in this disease. The *KCNJ5* gene encodes an integral membrane protein which belongs to one of seven subfamilies of inward-rectifier K^+^ channel proteins, called K^+^ channel subfamily J (Kir3.4), that regulate cell membrane potential [22]. Inward rectifier K^+^ channels are characterized by a greater tendency to allow K^+^ to flow into the cell rather than out of it. Their voltage dependence is regulated by the concentration of extracellular K^+^; as external K^+^ is raised, the voltage range of the channel opening shifts to more positive voltages [33].

Mutations are all located near to or within the GIRK4 channel selectivity filter and lead to loss of K^+^ selectivity with increased sodium (Na^+^) conductance into the cytoplasm. Increased cellular influx of Na^+^ depolarizes the plasma membrane and activates voltage-dependent Ca^2+^ channels, leading to intracellular accumulation of Ca^2+^ and activation of Ca signaling (the main regulator of aldosterone production), which ultimately triggers aldosterone production [30].

Various germline mutations of *KCNJ5* have been described in families presenting with FA. Indeed, PA severity depends on the type of mutation, and the clinical features of the affected cases vary all along the PA spectrum from mild and treatment-responding forms to severe PA with progressive disease. Symptoms may even mimic diabetes insipidus or Cushing’s syndrome [34]. This variability seems to be dependent on the type of the grounding *KCNJ5* mutations, among other factors. Carriers of arginine, glycine and tyrosine, alanine, serine, and glutamine mutations (p.Gly151Arg, p.Thr158Ala, p.Ile157Ser and p.Glu145Gln) all show a severe PA phenotype with early resistant HT [35,36]. Carriers of the *KCNJ5* mutations p.Gly151Glu and p.Tyr152Cys show moderate FA, which may be diagnosed later in young adulthood, with or without hypokalemia, similar to FA type II [37]. Other infrequent germline alterations of *KCNJ5* (some of them de novo) have been described. The variants of p.Glu145Gln, p.Ar-g52His, p.Glu246Lys, p.Gly247Arg alter channel functionality and increase aldosterone production compared with the wild-type protein [22].

Progress in understanding the role of *KCNJ5* in excessive aldosterone secretion has also been determined from pharmacological evidence showing that mutated GIRK4 channels are less sensitive to tertiapine-Q, a selective wild-type inhibitor of GIRK4. In addition, mutated GIRK4 activity is blocked by the epithelial Na + channel inhibitor amiloride and even more potently by the phenylalkylamine L-type Ca channel blocker verapamil. Conversely, the dihydropyridine Ca channel blocker nifedipine is a weak mutated channel blocker [30]. This information has the potential to stimulate the search for and development of new and selective mutated GIRK4 channel blockers, which are predicted to be effective in patients with type III FA and in a large subset of patients with sporadic PA.

#### 2.1.4. FA Type IV

A new form of FA, FA type IV, was recently identified by whole-exome sequencing in 40 patients with hypertension and PA before the age of 10 years [38]. FA type IV is exceedingly rare, attributable to mutations in the *CACNA1H* gene located on chromosome 16p13 encoding T-type voltage-gated calcium channel (Cav3.2.) [23]. In addition, de novo mutations in a gene for *CACNA1D* can cause PA, and are also linked to childhood seizures and neurologic abnormalities including neurocognitive disorders, epilepsy, and autism (PASNA).

In the adrenal zona glomerulosa, the increase in cytosolic Ca^2+^ concentration stimulates the signaling pathway leading to the transfer of cholesterol, the first step of aldosterone biosynthesis [33]. *CACNA1H* is the second most expressed Ca channel gene in adrenal zona glomerulosa [23]. The T-type Ca channel belongs to the low-voltage activated group, gives rise to T-type Ca^2+^ currents and opens at quite negative potentials. This type of channel serves pacemaking functions in both central neurons and cardiac nodal cells and supports Ca^2+^ signaling in secretory cells and vascular smooth muscle.

Germline *CACNA1H* mutations have been associated with several diseases, including epilepsy, autism and amyotrophic lateral sclerosis [22]. Daniil et al. identified de novo germline methionine (Met) and valine (Val) mutations affecting the *CACNA1H* gene (p.Met1549Val) [39]. The p.Met1549Val mutation changes the functional properties of the channel, facilitating its opening and impairing inactivation. This, like the other genetic abnormalities, leads to increased intracellular Ca^2+^ concentration and activation of the Ca^2+^ signaling pathway [14].

It is suggested that other factors, such as genetic modifiers, somatic mosaicism or the age of the patient, could restrain the gene defect [35]. This fact could explain the differences in disease presentation among patients with FA type IV, where some family members with mutations in p.Met1549 are affected with resistant HT and PA in early childhood while others display a milder or even normotensive phenotype [39]. In their study, Daniil et al. reported four additional variants of Met and isoleucine (Ile), serine (Ser), leucine (Leu), proline (Pro), glutamine (Glu): p.Met 1549Ile, p.Ser196Leu, p.Pro2083Leu, p.Val1951Glu. In addition, p.Val1951Glu was identified in a patient affected with an APA, with no familial history of PA available. All mutations altered Cav3.2 function and enhanced aldosterone production to a greater or a lesser degree [39]. These data may lead to conclusions that the phenotype seems to show incomplete penetrance, with some mutation carriers having no history of PA or HT or renin levels at the upper limit of the normal range despite mutations that lead to full-blown PA.

PASNA (PA associated with seizure and neurological abnormalities) is a clinical syndrome characterized by PA and neurological manifestations. It is caused by gain-of-function mutations in the *CACNA1D* gene located on chromosome 3p14.3, coding for the α1D subunit of an L-type voltage-gated Ca channel (Cav 1.3) that is expressed in the adrenal gland [23]. Scholl et al. reported two pediatric patients affected by PASNA syndrome due to de novo *CACNA1D* germline mutations (p.Gly403Asp and p.Ile770Met) [40]. The index cases presented with early-onset severe HT, hypokalemia, and neurological abnormalities (seizures, cerebral palsy and possibly autism spectrum disorder). It is presumed that the *CACNA1D* mutation may provide the genetic basis for the classification of another type of FA.

Additionally, a new missense *CACNA1D* germline mutation (p.Val104Leu) was identified in a patient affected by autism and epilepsy with a phenotype partially overlapping with that observed in patients with PASNA syndrome [41]. Interestingly, it was also found that that mutant Ca^2+^ channels retained full sensitivity toward the clinically available Ca channel blocker isradipine. This finding may encourage experimental therapy with available channel-blockers for this mutation.

Notably, the presence of a specific mutation can determine the treatment of choice. The T-type Ca channel blocker treatment seems to abrogate the aberrant *CYP11B2* activation and aldosterone production among patients with overexpressing Cav3.2 p.Met-1549Val mutant channels, which indicates that drugs of this class could be beneficial in the treatment of patients with FA type IV [42]. Ca channel blockers could target not only *CACNA1H* but also both *CACNA1H* and the L-type calcium channel *CACNA1D* [40]. In one of the two index patients, blood pressure was successfully controlled by the Ca channel blocker amlodipine, raising the possibility that calcium channel blockers might represent a specific treatment for individuals affected by APAs carrying a *CACNA1D* somatic mutation.

### 2.2. Mutations in APA and BAH

Numerous studies have now been performed examining tissue from surgically resected APAs to identify mutations characteristic of various disorders, including PA. Gain-of-function mutations in different genes, coding for cation channels (*KCNJ5*, *CACNA1D*, *CACNA1H*) and ATPases (*ATP1A1* and *ATP2B3*), regulating intracellular ion homeostasis, and plasma membrane potential have been described in in nearly 88% of sporadic APAs [43]. The most commonly identified mutations are *KCNJ5* (43%), *CACNA1D* (21%), and *ATP1A1* (17%) (Table 3) [30]. While rare, somatic mutations in the genes *CACNA1H* and *CLCN2* also have also been identified in APA [44,45].

#### 2.2.1. KCNJ5

In about 50% of patients with aldosterone-producing APA, somatic mutations of the *KCNJ5* gene have been demonstrated, as a result of which K + channels are less selective, allowing inflow of Na^+^ ions and then Ca^2+^ ions into the cells of the adrenal glomerular layer, which increases the synthesis of aldosterone [46]. Moreover, a recent meta-analysis including 13 studies for a total of 1636 patients presented the overall prevalence of *KCNJ5* mutations as 43%, with higher prevalence in East Asian populations compared to Western populations [47]. The studies showed that mutated *KCNJ5* exhibits a different pharmacological profile compared to the wild type; for example the Arg52His and Glu246Lys mutations caused cell depolarization and increased aldosterone production in vitro, in contrast to the Gly247Arg mutation which had no functional effect [48].

Female patients are more often carriers of these mutations than males [47]. In APA patients with *KCNJ5* mutations, the symptoms of hyperaldosteronism are particularly severe; PA occurs in the younger age and is associated with severe HT, aldosterone secretion is much higher, and K + concentration is significantly reduced. Patients also demonstrate high rates of clinical cure following adrenalectomy [33,49]. Carries of this gene had a larger APA tumor size than people without this mutation [50].

Although Ca channels are expressed in the adrenal tissue, it remains unclear whether PA driven by these mutations is particularly susceptible to treatment with Ca channel blockers. The presence of specific mutations may influence the choice of therapeutic treatment. Mutated *KCNJ5* exhibits a pharmacological profile that differs from that of wild-type channels. The mutated *KCNJ5* is less barrium (Ba^2+^)- and tertiapin-Q sensitive, but is inhibited by blockers of Na^+^ and Ca^2+^-transporting proteins, such as verapamil and amiloride [51]. The calcium channel blocker verapamil strongly inhibits the p.Leu168Arg mutant channel, suggesting a potential therapeutic use of this drug. Testing for these mutations is not currently part of standard clinical practice, and how this knowledge will practically influence routine clinical practice remains to be determined.

In a recent development, macrolide antibiotics such as roxithromycin and idremcinal were identified as selective inhibitors of GIRK4 p.Gly151Arg and p.Leu168Arg, with ablation of *CYP11B2* gene expression and consequent inhibition of aldosterone production [52]. Further exploration of the macrolides showed that *KCNJ5* was similarly selectively inhibited by idremcinal, a macrolide motilin receptor agonist, and by synthesized macrolide derivatives lacking antibiotic or motilide activity. This clinical trial showed that macrolide-derived selective *KCNJ5* inhibitors have the potential to advance the diagnosis and treatment of APAs carrying *KCNJ5* mutations.

#### 2.2.2. CACNA1D

Currently, a total of more than 31 different *CACNA1D* mutations have been reported, accounting for 9.3% of the sporadic APAs [48]. In addition to *KCNJ5*, *CACNA1D* is the gene with the second highest mutation burden in APAs. Accordingly, somatic *CACNA1D* mutations are the most frequent genetic alteration in *CYP11B2*-positive cortical micronodules in cross-sectional image-negative PA [53]. APAs carrying *CACNA1D* mutations are composed mainly of zona glomerulosa-like cells, and are smaller compared with those with *KCNJ5* or no mutations [39]. When using electrophysiology, both groups demonstrate that the somatic mutations discovered cause an increase in Ca^2+^ permeability, which can be inferred to directly lead to increased aldosterone production and proliferation [49].

The mutation of *CACNA* gene may have implication for therapy. It has been shown that in some patients carrying *CACNA1D* mutations, blood pressure was successfully controlled by the Ca channel blocker amlodipine, raising the possibility that calcium channel blockers might represent a specific treatment for individuals affected by APAs carrying a *CACNA1D* somatic mutation [40].

#### 2.2.3. ATP1A1 and ATP2B3

Beuschlein et al. identified the presence of somatic mutations in other genes, *ATP1A1* and *ATP2B3* [54]. The *ATP1A1* gene is located on chromosome 1p21 and encodes ATPase Na^+^/K^+^ transporting subunit alpha 1. Na^+^/K^+^-ATPase is an integral membrane protein responsible for establishing and maintaining the electrochemical gradients of Na^+^ and K^+^ across the plasma membrane [46]. These gradients are essential for osmoregulation, for Na-coupled transport of a variety of organic and inorganic molecules, and for the electrical excitability of nerve and muscle. Mutated ATPase has a lower affinity to K^+^ ions. It passively transports Na^+^ and K^+^ into the cell, which results in the depolarization of the membrane, opening of the voltage-gated Ca^2+^ channels and an increase in intracellular Ca^2+^ concentration, all without angiotensin II action, and as a consequence, an upregulation of aldosterone production [55].

The *ATP2B3* gene is located on chromosome X and encodes ATPase plasma membrane Ca^2+^ transporting 3. This magnesium-dependent enzyme catalyzes the hydrolysis of ATP coupled with the transport of CA^2+^ out of the cell. These enzymes remove Ca^2+^ from eukaryotic cells against very large concentration gradients, and play a critical role in intracellular Ca^2+^ homeostasis [54].

In a collection of 308 APAs, Beushlein et al. found sixteen (5.2%) somatic mutations in *ATP1A1* and five (1.6%) in *ATP2B3* [54]. The first mutation described by Beuschlein et al. was related to the segment M1 of ATPase, p.Leu104Arg (L104R), and the second to the segment M4, p.Val332Gly (V332G); a deletion in p.Phe100_Leu104del was also identified. Mutation-positive cases showed male dominance, increased plasma aldosterone concentrations and lower potassium concentrations compared with mutation-negative cases. It has been observed that *ATP1A1* and *ATP2B3* mutants had a higher membrane level of depolarization than cells without mutation. Both Na^+^/K^+^ ATPase and Ca^2+^ ATPase mutations occurred more frequently in male than in female patients [54]. The general prevalence of these mutations in patients with APAs was estimated to be about 3.0%. Thus far, further point-mutations in *ATP1A1* (p.Gly99Arg, p.GluGluThrAla963Ser, p.Leu425_Val426del, p.Val426_pVal427del.27) and *ATPB3* (c.1281_1286delGGCTGT, p.Arg428-Val429del, V426G_V427Q_A428_L433del, p.Val424_Leu425del.22,26,29,30) have been identified [46].

In addition, gene variants in *ATP2B4* were identified in a cohort of patients with BHA [56]. This gene encodes the plasma membrane calcium ATPase isoform 4. Endogenous ATP2B4 expression was characterized in adrenal tissue, and the gene variants were functionally analyzed for effects on aldosterone synthase (*CYP11B2*) expression and steroid production in basal and agonist-stimulated conditions. However, the study of Hattangady et al. did not confirm a pathogenic role for *ATP2B4* variants in BHA [57].

#### 2.2.4. ARMC5

The *ARMC5* gene maps on 16p11 and encodes an apoptosis regulator that belongs to the ARM (armadillo/beta-catenin-like repeat) superfamily. The ARM repeat is an approximately 40 amino acid-long tandemly repeated sequence motif. This repeat is implicated in the mediation of protein–protein interactions. Mutated *ARMC5* promotes cell survival and cortisol production in vitro [58]. Inactivation of *ARMC5* is associated with decreased expression of *CYP11B2*, the gene encoding aldosterone synthase enzyme. This is consistent with the effects of *ARMC5* on other steroidogenic enzymes.

Mutations in this gene are associated with primary bilateral macronodular adrenal hyperplasia, which is also known as ACTH-independent macronodular adrenal hyperplasia [58]. Zilbermint et al. investigated a cohort of 56 patients with PA for *ARMC5* defects [59]. It was found that 10.7% of the cohort with PA had predicted damaging *ARMC5* gene defects, which may be a novel cause of PA. Interestingly, the study showed that ethnic background may play a role in this condition, given that 100% of affected subjects were African Americans. However, the connection between *ARMC5* mutations and PA remains to be better understood. Mulatero et al. investigated the presence of germline *ARMC5* mutations in a group of 39 PA patients who had bilateral computed tomography-detectable adrenal alterations [60]. They concluded that *ARMC5* mutations are not present in a fairly large series of Caucasian patients with PA associated with bilateral adrenal lesions.

#### 2.2.5. CTNNB1

APAs can in rare cases also carry gain-of-function mutations in the *CTNNB1* gene, which is located on chromosome 3, encoding for β-catenin. β-catenin is part of a complex of proteins that constitute adherens junctions and the effector of the canonical Wnt signaling pathway [48]. This pathway controls key developmental gene expression programs, and constitutive activation of this pathway is involved in the pathogenesis of many human cancers. In APAs carrying *CTNNB1* mutations, the nuclear and/or cytoplasmic accumulation of active β-catenin protein has been shown to be increased especially for female patients, and the accumulation of β-catenin protein may be involved in APA proliferation and anti-apoptotic processes [61].

Constitutive activation of *CTNNB1* in mice has been demonstrated to trigger adrenal hyperplasia, aldosteronism and, at advanced age, malignancy [62]. Activating somatic mutations in the *CTNNB1* gene have been reported at a similar prevalence in both APAs (27% of 26 tumors, with a higher proportion in nonfunctioning adenomas compared with cortisol-producing adenomas) and malignant adrenocortical tumors (31% of 13 adrenocortical carcinomas) [63]. Somatic mutations in *CTNNB1* have been identified in around 3% of sporadic APAs and have been associated with female gender and relatively large adenomas [49]. Akerstrom et al. screened for *CTNNB1* mutations in a cohort of 198 APAs and provided compelling evidence that aberrant Wnt signaling caused by mutations in *CTNNB1* occurs in APAs [62]. In the study, somatic *CTNNB1* mutations were detected in 5.1% of the tumors. The mutations were associated with stabilized β-catenin, suggesting that activation of Wnt signaling and increased *CYP11B2* protein expression leads to increased aldosterone production in tumor tissue. The exact mechanisms underlying CTNNB1-mediated APA formation have not yet been clarified.

#### 2.2.6. mTORC1, EVs and Other Area of Active Investigation

The involvement of the mammalian target of rapamycin complex 1 (mTORC1) pathway in PA has been investigated. Swierczynska et al. identified mTOR signaling to be the most deregulated pathway in adenomas [64]. mTORC1 is activated in a subset of patients with PA, and treatment with the pathway inhibitor everolimus significantly decreased blood pressure and increased renin levels; however, it did not decrease aldosterone levels significantly among patients with PA. Prominent reduction of aldosterone levels upon everolimus treatment was observed in four patients [65].

Other area of active investigation in PA include exploration of the involvement of extracellular vehicles (EVs), which reflect endothelial cell activity and represent one of the mediators of endothelial dysfunction in PA patients [66]. One analysis demonstrated that patients with PA present a higher absolute number of endothelial-derived EVs compared with both patients with primary HT and normotensive controls. A microarray platform was validated by quantitative real-time polymerase chain reaction on four genes (*CASP1*, *EDN1*, *F2R*, and *HMOX1*) involved in apoptosis, inflammation, and endothelial dysfunction. The CASP1 gene encodes caspase 1, a protein which is a member of the cysteine-aspartic acid protease (caspase) family. Sequential activation of caspases plays a central role in the execution phase of cell apoptosis. The *EDN1* gene encodes endothelin 1, a peptide which is a potent vasoconstrictor, the cognate receptors of which are therapeutic targets in the treatment of pulmonary arterial hypertension. Aberrant expression of this gene may promote tumorigenesis. The *F2R* gene encodes coagulation factor II receptor that is involved in the regulation of thrombotic response, and may play a role in platelet activation as well as in vascular development. The *HMOX1* gene encodes heme oxygenase 1, an essential enzyme in heme catabolism; this enzyme contributes significantly to the apoptosis process. Through EV mRNA profiling, the *EDN1* gene was expressed only in patients with PA. Additionally, after unilateral adrenalectomy, the EV number and expression of *CASP1* and *EDN1* significantly decreased in the analyzed group of patients, which may confirm the role of circulating EVs in vascular dysfunction in patients with PA.

### 2.3. Aldosterone-Producing Cell Clusters

Next-generation sequencing (NGS) has revealed recurrent somatic mutations in aldosterone-driving genes in APA. By applying CYP11B2 immunohistochemistry and NGS to adrenal glands from normal subjects and PA patients, Omata et al. showed that CYP11B2-positive cells made small clusters, termed aldosterone-producing cell clusters (APCC), beneath the adrenal capsule and harbored somatic mutations in mutated genes in APA [67]. APCCs can be found in normotensive and hypertensive adrenal glands, and appear to exist in greater quantities with older age; APCCs frequently persist in the adrenals with an aldosterone-producing adenoma, suggesting autonomous CYP11B2 expression in these cells as well [68].

This was confirmed by finding known mutations that drive aldosterone production in adenomas in the APCCs of clinically normal people. Unilateral aldosteronism may also be due to multiple CYP11B2-expressing nodules of various sizes, or to a continuous band of hyperplastic zona glomerulosa cells expressing CYP11B2 [68]. Use of CYP11B2 antibodies to identify areas for sequencing has greatly facilitated the detection of aldosterone-driving mutations.

Expression analysis of these APCCs has revealed known aldosterone driver mutations including *CACNA1D* and *ATP1A1*, found in 35% of APCCs (but not in *KCNJ5*), supporting the hypothesis that they might display autonomous aldosterone overproduction and progress to overt PA over time [69]. Additionally, clinical correlates indicate that APCC number and size increases with age, and that this is associated with a pattern of decreased normal zona glomerulosa CYP11B2 expression and increased APCC expression [70]. This process might account for the age-related changes in renin and aldosterone physiology, and thus provide a potential explanation for age-related cardiovascular risk. It is certainly worth considering whether APCCs might add to the pathogenesis of HT.

## 3. Conclusions

The topical diagnostics and determination of the different variants of PA are highly sophisticated. It should be emphasized that the complexity of PA lies in the many morphological and genetic subtypes of the pathology and the peculiarities of the morphological structure of APAs, in particular the abnormal presence of cells of the zona fasiculata and glomerulosa. The relatively high prevalence of PA and the high rate of cardiovascular complications make it important for the clinician not to miss the diagnosis of PA.

The last few years have witnessed major advances in the field of both sporadic and familial PA. Three novel familial forms have been characterized, and somatic mutations altering intracellular ion homeostasis have been found to drive aldosterone overproduction in around 60% of sporadic APAs. The new insights into the molecular mechanisms underlying PA may be associated with potential implications for diagnosis and therapy. The evaluation of novel therapies might provide greater efficacy and safety; the drugs that target known genetic mechanisms of aldosterone secretion and are already effective in treating and controlling the symptoms of the disease. Despite the broad portfolio of somatic mutations and their mechanistic implications discussed above, genetic mutations are currently not exploited for diagnostic or therapeutic purposes. The deleterious effect of these mutations is still quite unclear, as most variants are not predicted to likely be pathogenic. Thus, the functions of the affected genes provide important cues to the underlying mechanisms of HT and will add opportunities for precision treatments in the future. In light of these recent findings, further studies must confirm or refute the possible role of these mutations in the etiology of PA.

## Figures and Tables

**Table 1 biomedicines-09-01708-t001:** Recommendation for PA screening [10].

Patients with sustained blood pressure above 160–179/100–109 mmHg) or systolic blood pressure above >180/110 mmHg
Patients with resistant HT (blood pressure not controlled by three conventional drugs including a diuretic) or controlled BP (<140/90 mmHg) on four or more antihypertensive drugs
Patients with HT and spontaneous or diuretic-induced hypokalemia
Patients with HT and an adrenal incidentaloma
Patients with HT and a family history of early-onset hypertension or cerebrovascular accident at a young age (<40 years)
All first-degree relatives of patients with PA
Patients with HT and sleep apnea

**Table 2 biomedicines-09-01708-t002:** Characteristics of familiar hyperaldosteronism.

	FA Type I	FA Type II	FA Type III	FA Type IV
Transmission	AD	AD	AD	AD or de novo
Gene	Chimeric *CYP11B1/B2*	*CLCN2*	*KCNJ5*	*CACNA1H*
Protein	Aldosterone synthase	Not known	GIRK4	Cav3.2
Biological effect	Aldosterone synthase inducible by ACTH stimulation	Increased Cl- efflux	Reduced K^+^ selectivity and increased Na^+^ influx	Increased Ca^2+^ entry
Age at symptom onset	Often <20 years	Variable, often <20 years	Variable, often <20 years	<10 years
PA in relatives	Yes	Yes	<20 years (variable in moderate forms)	Yes/no
Particular characteristics	Familiar history of stroke <40 years, hybrid steroids in urine, BAH	APA/BAH	BAH	Not described
Severity of hypertension	Normal or resistant	Normal or resistant	Extremely severe	Normal or severe
Hypokalemia	Yes/no	Yes/no	Very often in severe forms	Yes/no
Aldosterone level	Normal or high	Normal or high	Very high	Normal or high
Therapy	Glucocorticoids, MRAs	MRAs, adrenalectomy	MRAs, bilateral adrenalectomy	Not standardized, likely positive effect ofcalcium chanel blockers

AD—autosomal dominant; MRAs—mineralocorticoid receptor agonists.

**Table 3 biomedicines-09-01708-t003:** Most common mutations detected in APA and BAH.

Genetic Variant	Prevalence (%)	Affected Protein
*KCNJ5*	43%	GIRK4
*CACNA1D*	21%	Cav 1.3
*ATP1A1*	17%	ATPase Na^+^/K^+^ transporting subunit alpha 1
*ATP2B3*	unknown	ATPase plasma membrane Ca^2+^ transporting 3
*CACNA1H*	unknown	Cav 3.2
*ARMC5*	unknown	Aarmadillo repeat containing 5
*CTNNB1*	unknown	Catenin beta 1

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
