# Peer review of "Progress on Genetic Basis of Primary Aldosteronism"

_biomedicines, 2021, doi:10.3390/biomedicines9111708_

Round 1
Reviewer 1 Report
This manuscript is a review on the progress that has been done in the area of primary aldosteronism. The manuscript is thoroughly referenced but would have benefitted from more careful reading.
- Introduction, paragraph 2, line 1: glomerulosa (and not glomerular)
- Introduction, paragraph 2, line 3: angiotensin II
- Introduction, paragraph 2, line 5: comma (,) after hyponatremia
- Albeit rare, somatic mutations in the genes CACNA1H and CLCN2 have been identified in APA. Please reference the appropriate articles (Nanba et al, Hypertension 2020; Rege et al, JCEM 2020)
Author Response
I am grateful for the review, I have included all the corrections in the article.
Best regards,
Izabela Karwacka
Reviewer 2 Report
This review by Karwacka et al is highly comprehensive about Primary Aldosteronism. The manuscript is well written.
I think it deserves to be accepted for publication.
Author Response
I am very grateful for the review.
Best regards,
Izabela Karwacka
Reviewer 3 Report
This is a fine review of aldosteronism.
Three questions:
- You list numerous genetic mutations. Are some more common than others for adenomas vs hyperplasia?
- In the clinical section (page 3) you suggest to stop most medications for several weeks, replenish K, etc. Your suggestion, to this reviewer, seems "idealistic." In the real world if one detects SBP >160 and low K, it would appear that the diagnosis is self evident. Stopping HTN medications for several weeks would appear to be dangerous. Please comment.
- How do the mutations associate with hyperplasia and adenomas? The genetics describe ion channel problems but they do not explain how or why adenomas develop. Likewise, do channel mutations in other diseases lead to hyperplasia in other organs?
ENGLISH
- ABSTRACT "..higher rate of CVD." Change to "high".........."the higher the arterial pressure in patients, the higher..." remove "in patients"
- INTRO - "..... hyponatremia adrenocorticototropic hormone.." put a comma between the 2 words................"normovolemia (in the initial phase.." is a closing parenthesis missing?
- FA type III - "oksy" is "oxy"
- KCNJ5 - second paragraph - "carries" is "carriers"
Author Response
Question 1 and 3:
I discuss genetic mutations, their influence on the mechanisms of PA formation and the incidence of adenoma vs hyperplasia in detail in part of FA (e.g. in Table 2). I referred to this briefly in the APA section of the article, mainly to avoid duplicating information. Due to the fact that canalopathies are a new, rapidly developing field and, additionally, studies on relatively small groups of patients, the amount of data is very limited, if I had access to such information, they were taken into account.
Channel mutations lead to other diseases – for example ClC2 mutation leads to myotonia; ClC2 mutation in mice display severe testicular degeneration, azoospermia and early post- natal degeneration of the photoreceptors, which eventually entails a complete degeneration of the retina.
Question 2:
My suggestions are based on guidelines from endocrine societies and are indeed very difficult to implement in daily clinical work. Currently, it is believed that patients with PA can be diagnosed without modifying the treatment, because if PA has developed, the production of aldosterone has become autonomous and we should obtain test results typical for PA, regardless of the drugs used. However, this clinical practice has not yet been included in the guidelines for the management of patients with PA.